# The Accuracy of Lateral Cephalogram in Representing the Anterior Maxillary Dentoalveolar Position

**DOI:** 10.3390/diagnostics12081840

**Published:** 2022-07-30

**Authors:** Supontep Teerakanok, Chairat Charoemratrote, Pannapat Chanmanee

**Affiliations:** 1Periodontic Section, Department of Conservative Dentistry, Faculty of Dentistry, Prince of Songkla University, Hat Yai 90112, Thailand; supontep.t@psu.ac.th; 2Orthodontic Section, Department of Preventive Dentistry, Faculty of Dentistry, Prince of Songkla University, Hat Yai 90112, Thailand; metalbracket@hotmail.com

**Keywords:** accuracy, cephalometrics, CBCT, dentoalveolar position, error in diagnosis

## Abstract

Background: To evaluate the dentoalveolar position and root diameters of the maxillary incisors from cone beam computed tomograms (CBCT) compared with cephalometric tracings. Methods: A total of 64 sets of initial lateral cephalograms and CBCT images were enrolled. Measurements of dentoalveolar position included bone thicknesses and heights of alveolar, cortical, and cancellous bone. Root diameter and total root-bone thickness were also evaluated. All measurements were performed on cephalograms and CBCT images of the maxillary central incisor (U1CT) and maxillary lateral incisor (U2CT). The data were statistically analyzed using one-way ANOVA and Bonferroni tests (*p* < 0.01) to compare the cephalograms, U1CT, and U2CT. Results: The cephalograms presented thicker alveolar bone (labial: 0.20–0.67 mm, palatal: 0.41–0.60 mm; *p* < 0.001) and cortical bone (labial: 0.20–0.67 mm, palatal: 0.41–0.52 mm; *p* < 0.001) as well as higher alveolar crest (labial: 0.23–0.27 mm, palatal: 0.15–0.17 mm; *p* < 0.001) and cortical height (labial: 0.35–0.47 mm; *p* = 0.051, palatal: 0.14–0.18 mm; *p* < 0.001) than the CBCT images on both the labial and palatal sides, whereas palatal cancellous thickness was not significantly greater (*p* > 0.01). The cephalograms presented a greater total root-bone thicknesses (0.80–1.08 mm; *p* < 0.001), whereas the cephalograms traced thinner roots than the CBCT images (0.36–0.52 mm; *p* < 0.01). Conclusion: Routine lateral cephalograms are not suitable for alveolar bone evaluation in orthodontic treatment due to errors in representing dentoalveolar thicknesses and heights.

## 1. Introduction

The lateral cephalogram is a routine radiograph that orthodontists use for diagnosis and treatment planning. Cephalometric images present two-dimensional structures of the skeleton, teeth, and soft tissue involving relationships and deformities [1]. Moreover, cephalometric images are frequently used to determine the suitable position for maxillary incisor retraction or protraction [2]. Desirable orthodontic movement occurs when the tooth moves within the alveolar bone to reduce treatment complications such as gingival recession, external root resorption, bone dehiscence, and fenestration [3].

The maxillary anterior region is an important esthetic zone for orthodontic treatment planning [4]. When the maxillary incisors are retracted, the amount of palatal cancellous bone is important since it allows the root to move appropriately according to the treatment objective.

Alveolar bone serves as a labio-palatal envelope for tooth movement, especially labial bone composed of pure cortical bone. Therefore, the meticulous interpretation of the presenting bone surrounding tooth structures from radiographs should be done simultaneously before commencing orthodontic treatment [5]. Generally, lateral cephalometric tracings are used in treatment planning and only one central incisor is used to represent the four maxillary incisors to evaluate tooth position, inclination, and surrounding structures. Nevertheless, no previous literature has revealed a clear explanation of the interpretation of cephalometric tracing.

Nowadays, cone beam computed tomography (CBCT) is recommended to evaluate the teeth, roots, and alveolar bone structures in three dimensions because of its high accuracy and reliability [6]. Several studies showed excellent resolution of CBCT images to define the boundary and composition of alveolar, cortical, and cancellous bone [7,8,9]. Unfortunately, CBCT is not used routinely in orthodontic treatment due to its high radiation dose and costs compared to conventional radiographs.

The cephalometric interpretation of the dentoalveolar position and alveolar bone evaluation affect the accuracy of an orthodontic diagnosis and treatment plan. It should be done carefully before the commencement of orthodontic treatment. However, no previous study has revealed the limitations of lateral cephalogram in representing the anterior maxillary dentoalveolar position compared to CBCT images in different tooth types of maxillary incisors. Therefore, the objective of this study was to compare the results of CBCT images and lateral cephalograms in representing the anterior maxillary dentoalveolar position and root diameters of the maxillary central and lateral incisors.

## 2. Materials and Methods

### 2.1. Subject Selection

This study was carried out on cephalograms and CBCT images from pretreatment orthodontic records of patients who attended the orthodontic clinic at the Faculty of Dentistry, Prince of Songkla University from 2014 to 2020. The study protocol was approved by the Institutional Review Board for human patients (protocol EC6305-016) of the Faculty of Dentistry, Prince of Songkla University.

A total of 64 subjects (28 males, 36 females) with a mean age of 20.64 years (range 18–29) were recruited in this study. The inclusion criteria were as follows: (1) healthy adults aged 18 to 30 years; (2) well-aligned maxillary anterior teeth; (3) no periodontal diseases; (4) no history of facial or dental trauma; (5) no previous orthodontic or orthognathic surgery; (6) no significant medical illness related to bone metabolism; (7) no motion artifact or metal artifact at lower incisors area; and (8) good quality images and contrast resolution. The exclusion criteria were as follows: (1) rotation or crowding; (2) history of orthodontic treatment; and (3) previous surgery in the maxillary anterior region.

### 2.2. Lateral Cephalograms

Lateral cephalograms of all subjects were taken in the natural head position as the reference [10]. The same cephalostat and cephalometric machines were used for all subjects. The maxillary central incisor and underlying alveolar bone on each lateral cephalogram were traced digitally. All radiographs were digitized and analyzed by Dolphin Imaging^®^ (version 11.9; Dolphin Imaging, Chatsworth, CA, USA). On the lateral cephalograms, the measurements were converted to 100% magnification. The scale ruler on the cephalogram was used to perform the mathematical conversion by ImageJ software (version 1.53a, NIH, Bethesda, MD, USA).

The tooth’s long axis was used as a reference. Thicknesses including alveolar bone, cortical bone, cancellous bone on labial and palatal sides, root diameter, and total root-bone thickness of the cephalometric maxillary central incisor were measured perpendicular to the long axis at 3, 6, and 9 mm apical to the cementoenamel junction (CEJ) (Figure 1) [11]. Height including labial and palatal sides was measured parallel to the tooth axis. All cephalometric parameters were measured in millimeters with two significant digits by ImageJ software and designated as “Ceph”.

### 2.3. Cone Beam Computed Tomography (CBCT)

CBCT images of the maxillary incisors were scanned using (80 kV, 5 mA, 9.2 s exposure time, 0.125 mm voxel resolution, 80 × 80 mm field of view; Veraviewepocs J Morita Mfg. Corp., Fushimi-ku, Kyoto, Japan). CBCT data were reconstructed at 0.125 mm increments. Each CBCT radiograph was oriented along the tooth’s long axis of the root and the sagittal plane running transversely through the midpoint of the tooth axis (Figure 2A). The sagittal CBCT image was used for all measurements (Figure 2B) from the central and lateral incisors following the same vertical references as the lateral cephalograms (3, 6, and 9 mm apical to CEJ). The thickness and height measurements were in millimeters to the nearest two digits by i-Dixel One Volume Viewer software (J Morita Mfg. Corp., Fushimi-ku, Kyoto, Japan). The maxillary central incisor and lateral incisor were assigned the terms U1CT and U2CT, respectively, for the measured parameters of the CBCT radiographs.

### 2.4. Statistical Analyses

The Shapiro–Wilk test showed normally distributed variables; therefore, the differences between the three groups (Ceph, U1CT, and U2CT) were tested using one-way analysis of variance (ANOVA) followed by Bonferroni tests. All statistical analyses were performed using SPSS version 17 (SPSS, Chicago, IL, USA). The level of significance of all tests was set at *p* < 0.01.

### 2.5. Sample Size Calculation

The sample size calculation followed a study by Park et al. to detect a difference between conventional lateral cephalograms and the corresponding CBCT radiographs to provide power above 80% [12]. The calculation indicated that 64 subjects were required.

### 2.6. Quality Control

All measurements were performed by one examiner blinded to all subjects. Twenty-five randomly selected subjects were remeasured after an interval of two weeks to assess measurement error and reliability. Comparison between the first and second measurements using the independent t-test illustrated no significant differences between the two sets (*p* < 0.01). The intraclass correlation coefficient was higher than 0.92 for all measurements, which indicated excellent reliability (*p* < 0.05). No systematic error was observed for any variable in the paired t-test (*p* > 0.05). Random errors were estimated by the Dahlberg formula (ME^2^ = Σd^2^/2n) which varied from 0.09 mm to 0.11 mm for linear cephalometric measurements, and from 0.02 mm to 0.04 mm for linear CBCT measurements. These random errors were considered acceptable.

## 3. Results

### 3.1. Comparisons between Labial Ceph, U1CT, and U2CT

Comparisons between Ceph, U1CT, and U2CT of labial side were shown in Table 1. The Ceph labial alveolar bone thickness gradually increased toward the apical area, especially at the 9 mm level (1.32 mm) but the CT labial alveolar bone thicknesses for both U1CT and U2CT were almost constant at all levels. The Ceph showed a significantly thicker labial alveolar bone than U1CT and U2CT (*p* < 0.001) but no significant differences were found between U1CT and U2CT at all levels. The differences between Ceph-U1CT and Ceph-U2CT were 0.20 to 0.64 mm and 0.24 to 0.67 mm, respectively. The distance from the CEJ to the alveolar crest was the shortest in Ceph, whereas U1CT and U2CT were almost similar. The differences between Ceph-U1CT and Ceph-U2CT were 0.23 and 0.27 mm, respectively.

The Ceph labial cortical bone thickness gradually increased toward the apical area, whereas the CT labial cortical bone thicknesses for both U1CT and U2CT were almost constant at all levels, which was similar to the labial alveolar bone. Ceph showed thicker labial cortical bone than U1CT and U2CT (*p* < 0.001) but no significant difference was found between U1CT and U2CT. The differences between Ceph-U1CT and Ceph-U2CT were similar to the labial alveolar bone thickness. The cortical bone height of Ceph was the highest (10.51 mm), and the differences between Ceph-U1CT and Ceph-U2CT were 0.35 and 0.47 mm, respectively; however, statistically significant differences were found in all groups (*p* > 0.05). No cancellous bone was detected in the labial bone at the measured levels.

### 3.2. Comparisons between Palatal Ceph, U1CT, and U2CT

Comparisons between Ceph, U1CT, and U2CT of palatal side were shown in Table 2. The palatal alveolar bone thickness obviously increased toward the apical area in all groups. Ceph showed significantly thicker palatal alveolar bone than U1CT and U2CT (*p* < 0.001). The significant differences between Ceph-U1CT and Ceph-U2CT were 0.41 to 0.54 mm and 0.48 to 0.60 mm, respectively. However, no significant difference was found between U1CT and U2CT. The distance from the CEJ to the palatal alveolar crest was the shortest in Ceph (0.90 mm), whereas U1CT and U2CT had no significant differences. The differences between Ceph-U1CT and Ceph-U2CT were 0.15 and 0.17 mm, respectively.

The palatal cortical bone thickness gradually increased toward the apical area in all groups from 1.53 to 1.97 mm, 1.12 to 1.49 mm, and 1.05 to 1.45 mm for Ceph, U1CT, and U2CT, respectively. Ceph showed significantly thicker palatal cortical bone than U1CT and U2CT (*p* < 0.001). The significant differences between Ceph-U1CT and Ceph-U2CT were 0.41 to 0.48 mm and 0.48 to 0.52 mm, respectively. The difference between U1CT and U2CT was not statistically significant. The palatal cortical bone height was the highest in Ceph (3.05 mm), whereas U1CT and U2CT had no significant differences. Ceph was higher than U1CT and U2CT by 0.14 and 0.18 mm, respectively.

The palatal cancellous bone presented from 6 mm apical to the CEJ toward the root apex. The thicknesses distinctly increased toward the apical area in all groups from 0.49 to 1.12 mm, 0.41 to 1.06 mm, and 0.42 to 1.04 mm for Ceph, U1CT, and U2CT, respectively. However, no significant differences were found between Ceph, U1CT, and U2CT (*p* > 0.05).

### 3.3. Comparisons of Root Diameters and Total Root-Bone Thickness between Ceph, U1CT, and U2CT 

Comparisons of root diameters and total root-bone thickness between Ceph, U1CT, and U2CT were shown in Table 3. The root diameter gradually decreased from 3 mm apical to the CEJ toward the root apex in all groups. Ceph was significantly thinner than U1CT by 0.46 to 0.52 mm (*p* < 0.001), whereas Ceph was thinner than U2CT by 0.16 to 0.21 mm but no significant differences were found. Moreover, U1CT was significantly thicker than U2CT by 0.36 to 0.52 mm (*p* < 0.001).

The total root-bone thickness gradually increased from 3 mm apical to the CEJ toward the root apex in all groups. Ceph was significantly thicker than U1CT at the 9 mm level (0.80 mm) as well as significantly thicker than U2CT at all levels (0.78 to 1.08 mm), whereas U1CT was significantly thicker than U2CT only at levels 3 and 6 mm. The significant differences between U1CT-U2CT were 0.47–0.67 mm.

## 4. Discussion

This study recruited the records of lateral cephalograms and CBCT radiographs to compare the anterior maxillary dentoalveolar position and root and bone interpretation. Furthermore, comparisons of Ceph-U1CT and Ceph-U2CT were done to explore the difference between lateral cephalograms and CBCT images. A comparison between U1CT and U2CT was to assess the initial bone between tooth types when four maxillary incisors are retracted or protracted. The measurements and comparisons of alveolar, cortical, and cancellous bone compositions between lateral cephalograms and CBCT images are important since the morphology of the supporting bone determines the boundary of maxillary incisor movement [7]. Moreover, the root diameter measurement was introduced in the present study since an error in root tracing could affect the adjacent bone measurements.

Labial alveolar bone in Ceph was slightly thicker than U1CT or U2CT at the 3 and 6 mm levels (0.20 to 0.36 mm) but was obviously thicker at the 9 mm level (0.64 to 0.67 mm). The possible explanation came from two reasons. First, the thicker Ceph was the result of radiographic image magnification [13]. Second, tracing the line constructed from the CEJ got gradually thicker until it reached the anterior nasal spine following the anatomical landmarks of cephalometric tracing [14] such that Ceph was significantly thicker at the 9 mm level.

When the labial alveolar height was measured, the alveolar crest of Ceph was located closer to the CEJ than U1CT and U2CT. The possible explanation is the effect of cephalometric tracing that involved the adjacent proximal crestal bone that was located more incisal compared with the labial crestal bone [15]. Although the differences between the Ceph and CBCT images were 0.23–0.27 mm, it may not be clinically significant compared to the variation of normal healthy crestal height [16]. However, the actual labial bone height should be a concern when incisor protraction is introduced.

When the labial cortical bone was measured, the thickness had the same explanation as the labial alveolar bone thicknesses because the pure cortical bone was found at all measured levels. The height of the labial cortical bone of Ceph was higher thanU1CT and U2CT, which resulted from two reasons: (1) the more incisal tracing of the alveolar crest compared to CBCT and (2) the interface of the cortical and cancellous bone from Ceph was mostly detected apically. Cancellous bone was found on the labial bone at the measured levels because the pure cortical bone was found on the labial plate. When pure cortical bone is detected on all labial plates, a light force should be applied to generate the desirable bone remodeling [17].

The palatal alveolar bone was obviously thicker toward the apical area in Ceph and CBCT images because of the normal architecture of the healthy palatal bone [18]. Ceph showed a significantly thicker palatal alveolar bone (0.4–0.6 mm) than the CBCT images from the greater magnification of the cephalogram which affects clinical consideration when upper incisor retraction is planned. The palatal alveolar crest of Ceph was more incised by about 0.2 mm compared to the CBCT images. The explanation is the same as for the labial bone. The palatal alveolar crest was more apical than the adjacent proximal crestal bone. Although differences of 0.2 mm are not clinically significant, this information concealed the actual bone height that orthodontists need to be aware of when upper incisor retraction is planned. Furthermore, the amount of force and the bracket system affect incisor and root movement [19].

The palatal cortical bone was gradually thicker in the apical direction in both the Ceph and CBCT images, which was similar to previous literature [7]. The thicker palatal cortical bone of Ceph compared to the CBCT images was the result of greater magnification. The thicker palatal cortical bone in Ceph was 0.4 to 0.5 mm, which would be a concern when uncontrolled tipping induces undermined resorption [20]. The palatal cortical bone height of Ceph was higher than the CBCT images by about 0.2 mm, which has the same explanation as the labial cortical bone (i.e., an incisally traced alveolar crest and the apically detected interface of the cortical and cancellous bone). The palatal cancellous bone of Ceph was thicker than the CBCT images by about 0.1 mm from greater magnification; however, the differences were very little and not significant.

A gradual thinning of the root apically from the CEJ relied on the normal root anatomy [21]. Ceph presented thinner roots than the CBCT images by 0.4 to 0.5 mm, whereas Ceph was thinner than U2CT by about 0.2 mm. Moreover, U1CT was thicker than U2CT by 0.4 to 0.5 mm because the maxillary central incisor presented a thicker root than the lateral incisor by 0.5 mm, which followed Lee’s study [22]. Total root-bone thickness between the Ceph and CBCT images revealed an actual cephalometric magnification that was 0.8–1.0 mm, whereas total root-bone thicknesses of U1CT and U2CT presented an overall root-bone thickness of both tooth types that was 0.5–0.6 mm.

This study provides comprehensive measurements of maxillary alveolar bone compositions of CBCT images compared to routine lateral cephalograms. A lateral cephalogram is the standard tool for orthodontic treatment decisions and planning in terms of dentoalveolar position, esthetic evaluation, and treatment options (extraction/non-extraction/orthognathic surgery) [23]. However, the differences observed between Ceph and CBCT in each measurement came from several factors such as radiographic magnification, anatomical geometry of a proximal bone and anterior nasal spine, and errors in root tracing. Therefore, that cephalometric tracing is not suitable for alveolar bone evaluation in orthodontic treatment.

Errors in routine cephalometric tracing can come from both obtaining the cephalogram (due to radiographic enlargement or distortion) and landmark identification from poor image quality or lack of experience. The accuracy or error of a lateral cephalogram in representing the anterior maxillary dentoalveolar position in this study was observed in the different values between the measurements obtained from the cephalograms and CBCT images. The error in the lateral cephalogram in detecting bone thickness was 0.5 mm whereas errors in bone heights were 0.2–0.3 mm, and the error in root thickness was 0.5 mm.

The results of this study may be applied only to the maxillary incisors. Further investigations are recommended into the differences of cephalograms and CBCT to detect and evaluate dentoalveolar positions and root diameters of all dentition areas. In particular, the mandibular incisors are important for orthodontic tooth movement, and periodontal sequalae are prone to occur following orthodontic treatment in different types of craniofacial skeletal patterns [24,25].

## 5. Conclusions

Ceph presented thicker bone than the CBCT images due to cephalometric magnification and thinner root tracing. The differences are 0.2–0.6 mm for the labial side and 0.4–0.6 mm for the palatal side.

Ceph presented higher bone height than the CBCT images from the more incisal cephalometric tracing. The differences were about 0.3 mm for the labial side and about 0.2 mm for the palatal side.

Routine cephalometric tracing presented some errors in representing a dentoalveolar position in both the thicknesses and heights.

## Figures and Tables

**Figure 1 diagnostics-12-01840-f001:**
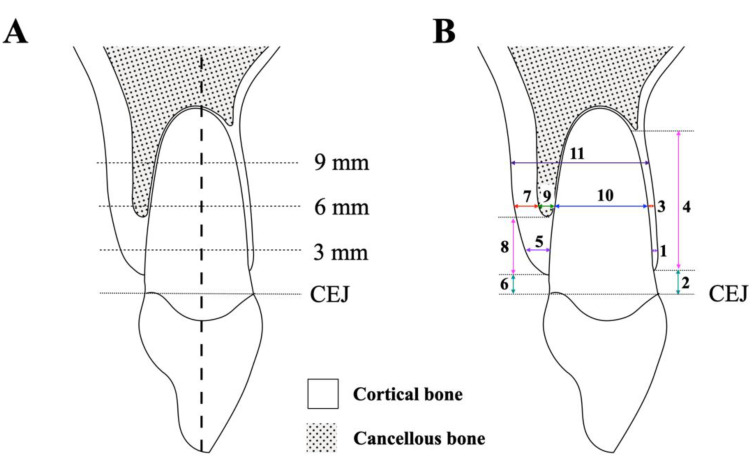
**Cephalometric measurements**. (**A**) The vertical references: the measurements perpendicular to the tooth’s long axis at 3, 6, and 9 mm apical to the CEJ. (**B**) Cephalometric measurements of (1) to (4) on labial side: (1) labial alveolar bone thickness, (2) labial alveolar bone height, (3) labial cortical bone thickness, (4) labial cortical bone height. Cephalometric parameters of (5) to (9) on palatal side: (5) palatal alveolar bone thickness, (6) palatal alveolar bone height, (7) palatal cortical bone thickness, (8) palatal cortical bone height, (9) palatal cancellous bone thickness, (10) root diameter, and (11) total root-bone thickness.

**Figure 2 diagnostics-12-01840-f002:**
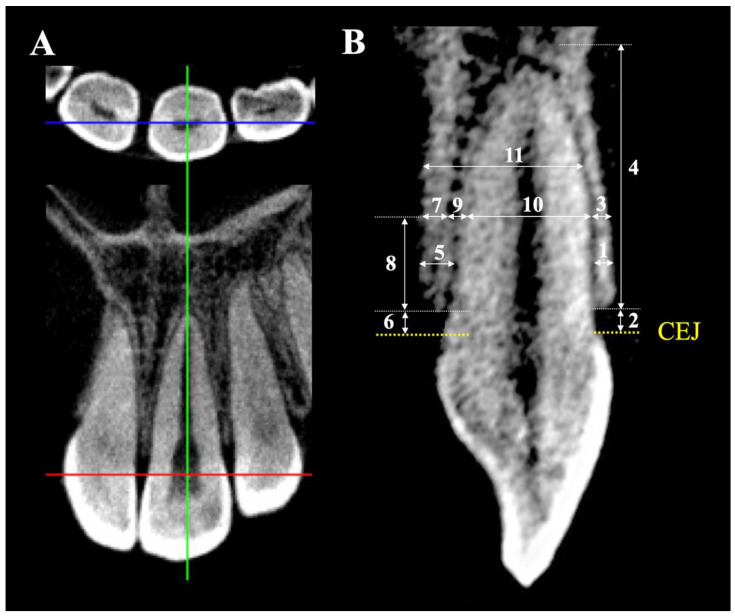
**CBCT measurements**. (**A**) Tooth orientation of CBCT: the sagittal plane running transversely through the midpoint of the tooth’s long axis. (**B**) CBCT measurements of (1) to (4) on labial side: (1) labial alveolar bone thickness, (2) labial alveolar bone height, (3) labial cortical bone thickness, (4) labial cortical bone height. CBCT measurements of (5) to (9) on palatal side: (5) palatal alveolar bone thickness, (6) palatal alveolar bone height, (7) palatal cortical bone thickness, (8) palatal cortical bone height, (9) palatal cancellous bone thickness, (10) root diameter, and (11) total root-bone thickness.

**Table 1 diagnostics-12-01840-t001:** Comparisons of labial alveolar bone between Ceph, U1CT, and U2CT.

Maxillary Teeth (*n* = 64)	Labial Side						
	Ceph	U1CT	U2CT	ANOVA *p*-Value	Differences		
	(C)	(U1)	(U2)		C-U1	C-U2	U1-U2
**1) Alveolar bone thickness**							
3 mm apical to CEJ	0.81 ± 0.15	0.61 ± 0.12	0.57 ± 0.09	<0.001	0.20 **	0.24 **	0.04
6 mm apical to CEJ	0.95 ± 0.21	0.64 ± 0.11	0.59 ± 0.10	<0.001	0.31 **	0.36 **	0.05
9 mm apical to CEJ	1.32 ± 0.19	0.68 ± 0.11	0.65 ± 0.14	<0.001	0.64 **	0.67 **	0.03
**2) Alveolar bone height (CEJ to alveolar crest)**	1.38 ± 0.27	1.61 ± 0.20	1.65 ± 0.17	<0.001	0.23 **	0.27 **	0.04
**3) Cortical bone thickness**							
3 mm apical to CEJ	0.81 ± 0.15	0.61 ± 0.12	0.57 ± 0.09	<0.001	0.20 **	0.24 **	0.04
6 mm apical to CEJ	0.95 ± 0.21	0.64 ± 0.11	0.59 ± 0.10	<0.001	0.31 **	0.36 **	0.05
9 mm apical to CEJ	1.32 ± 0.19	0.68 ± 0.11	0.65 ± 0.14	<0.001	0.64 **	0.67 **	0.03
**4) Cortical bone height**	10.51 ± 1.07	10.16 ± 0.96	10.04 ± 0.95	0.051	0.35	0.47	0.12
**5) Cancellous bone thickness**							
3 mm apical to CEJ	-	-	-	N/A	-	-	-
6 mm apical to CEJ	-	-	-	N/A	-	-	-
9 mm apical to CEJ	-	-	-	N/A	-	-	-

Differences between groups were tested by ANOVA and Bonferroni test. ** *p* < 0.001.

**Table 2 diagnostics-12-01840-t002:** Comparisons of palatal alveolar bone between Ceph, U1CT, and U2CT.

Maxillary Teeth (*n* = 64)	Palatal Side						
	Ceph	U1CT	U2CT	ANOVA *p*-Value	Differences		
	(C)	(U1)	(U2)		C-U1	C-U2	U1-U2
**1) Alveolar bone thickness**							
3 mm apical to CEJ	1.53 ± 0.44	1.12 ± 0.38	1.05 ± 0.32	<0.001	0.41 **	0.48 **	0.07
6 mm apical to CEJ	2.24 ± 0.41	1.71 ± 0.39	1.66 ± 0.30	<0.001	0.53 **	0.58 **	0.05
9 mm apical to CEJ	3.09 ± 0.42	2.55 ± 0.39	2.49 ± 0.37	<0.001	0.54 **	0.60 **	0.06
**2) Alveolar bone height (CEJ to alveolar crest)**	0.90 ± 0.18	1.05 ± 0.15	1.07 ± 0.12	<0.001	0.15 **	0.17 **	0.02
**3) Cortical bone thickness**							
3 mm apical to CEJ	1.53 ± 0.44	1.12 ± 0.38	1.05 ± 0.32	<0.001	0.41 **	0.48 **	0.07
6 mm apical to CEJ	1.75 ± 0.46	1.30 ± 0.36	1.24 ± 0.33	<0.001	0.45 **	0.51 **	0.06
9 mm apical to CEJ	1.97 ± 0.44	1.49 ± 0.36	1.45 ± 0.34	<0.001	0.48 **	0.52 **	0.04
**4) Cortical bone height**	3.05 ± 0.22	2.91 ± 0.24	2.87 ± 0.21	<0.001	0.14 *	0.18 **	0.04
**5) Cancellous bone thickness**							
3 mm apical to CEJ	-	-	-	N/A	-	-	-
6 mm apical to CEJ	0.49 ± 0.18	0.41 ± 0.21	0.42 ± 0.20	0.078	0.08	0.07	0.01
9 mm apical to CEJ	1.12 ± 0.15	1.06 ± 0.34	1.04 ± 0.37	0.387	0.06	0.08	0.02

Differences between groups were tested by ANOVA and Bonferroni test. * *p* < 0.01, ** *p* < 0.001.

**Table 3 diagnostics-12-01840-t003:** Comparisons of root diameters and total root-bone thickness between Ceph, U1CT, and U2CT.

Maxillary Teeth (*n* = 64)	Ceph	U1CT	U2CT	ANOVA *p*-Value	Differences		
	(C)	(U1)	(U2)		C-U1	C-U2	U1-U2
**1) Root diameters**							
3 mm apical to CEJ	5.61 ± 0.68	6.07 ± 0.65	5.54 ± 0.60	<0.001	0.46 *	0.07	0.52 **
6 mm apical to CEJ	5.16 ± 0.55	5.68 ± 0.58	5.32 ± 0.80	<0.001	0.52 **	0.16	0.36 *
9 mm apical to CEJ	4.64 ± 0.63	5.02 ± 0.63	4.85 ± 0.79	0.018	0.38	0.21	0.17
**2) Total root-bone thickness**							
3 mm apical to CEJ	7.95 ± 0.81	7.80 ± 0.81	7.15 ± 0.60	<0.001	0.15	0.80 **	0.65 **
6 mm apical to CEJ	8.35 ± 0.57	8.04 ± 0.77	7.57 ± 0.70	<0.001	0.31	0.78 **	0.47 *
9 mm apical to CEJ	9.05 ± 0.75	8.25 ± 0.79	7.97 ± 0.91	<0.001	0.80 **	1.08 **	0.28

Differences between groups were tested by ANOVA and Bonferroni test. * *p* < 0.01, ** *p* < 0.001.

## Data Availability

The data presented in this study are available on request from the corresponding author.

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
