# Peer review of "The Accuracy of Lateral Cephalogram in Representing the Anterior Maxillary Dentoalveolar Position"

_diagnostics, 2022, doi:10.3390/diagnostics12081840_

Round 1
Reviewer 1 Report
Dear Authors,
I’ve extensively read the manuscript titled “The Accuracy of Lateral Cephalogram in Representing 2 the Anterior Maxillary Dentoalveolar Position”. The aim of this study is to use CFD to evaluate dentoalveolar position and root diameters of the maxillary incisors from cone-beam computed tomograms (CBCT) compared with cephalometric tracings. The methodology is appropriate and quite linear with recent evidences/ studies on this topic. I’ve not major concerns in this regard.
Some aspects must be improved before considering the study suitable for publication
- The study is well conducted; however, it should be tailored for orthodontists. I mean, in the introduction the authors have limited to mention the relevance of the study. A wide reasoning about orthodontic diagnosis and treatment plan strategies is fundamental
- Similarly, in the discussion section the authors have limited to mention the fact that cephalometric tracing is not the suitable choice for alveolar bone evaluation 273 in orthodontic treatment. Again, a wide reasoning about the consequences of their findings in terms of orthodontic treatment decision and planning (extraction/non extraction, the usage of inter-maxillary elastics etc) is expected.
- Why did the authors do not consider to investigate mandibular incisors area? This region is even more important for orthodontic purposes, even considering the periodontal sequalae of orthodontic treatment. At least, Authors should mention this aspect.
It should be interesting to investigate the ceph/CBCT differences in detecting evaluate dentoalveolar position and root diameters of the maxillary incisors (even mandibular) in different craniofacial skeletal pattern reporting (Lo Giudice A, Rustico L, Caprioglio A, Migliorati M, Nucera R. Evaluation of condylar cortical bone thickness in patient groups with different vertical facial dimensions using cone-beam computed tomography. Odontology 2020;108(4):669-675; Lo Giudice A, Caccianiga G, Crimi S, Cavallini C, Leonardi R. Frequency and type of ponticulus posticus in a longitudinal sample of nonorthodontically treated patients: relationship with gender, age, skeletal maturity, and skeletal malocclusion. Oral Surg Oral Med Oral Pathol Oral Radiol. 2018 Sep;126(3):291-297)
- In Figure 2 seems that the sagittal axis (green) is not centered and not parallel to th long axis of the tooth, while for this kind of measurements it is necessary to have consistency among measurements.
Author Response
Response to Reviewer 1 Comments
Point 1: I’ve extensively read the manuscript titled “The Accuracy of Lateral Cephalogram in Representing 2 the Anterior Maxillary Dentoalveolar Position”. The aim of this study is to use CFD to evaluate dentoalveolar position and root diameters of the maxillary incisors from cone-beam computed tomograms (CBCT) compared with cephalometric tracings. The methodology is appropriate and quite linear with recent evidences/ studies on this topic. I’ve not major concerns in this regard. Some aspects must be improved before considering the study suitable for publication.
Response 1: Thank you very much for your valuable consideration. We have made changes following your suggestions.
Point 2: The study is well conducted; however, it should be tailored for orthodontists. I mean, in the introduction the authors have limited to mention the relevance of the study. A wide reasoning about orthodontic diagnosis and treatment plan strategies is fundamental.
Response 2: Thank you very much for your suggestions. We agree with the reviewer that relevance of the study should be mentioned. A wide reasoning about orthodontic diagnosis and treatment plan strategies is very important. Therefore, we added the relevance of this study in the last paragraph of the Introduction.
Previous content:
Introduction
No previous study revealed the limitations of lateral cephalogram in representing the anterior maxillary dentoalveolar position compared to CBCT images in different tooth types of maxillary incisors. Therefore, the objective of this study was to compare the results of CBCT images and the lateral cephalogram in representing the anterior maxillary dentoalveolar position and root diameters of the maxillary central and lateral incisors.
Amended content:
Introduction;
Cephalometric interpretation of dentoalveolar position and alveolar bone evaluation affect the accuracy of an orthodontic diagnosis and treatment plan. It should be done carefully before commencement of the orthodontic treatment. However, no previous study has revealed the limitations of lateral cephalogram in representing the anterior maxillary dentoalveolar position compared to CBCT images in different tooth types of maxillary incisors. Therefore, the objective of this study was to compare the results of CBCT images and lateral cephalograms in representing the anterior maxillary dentoalveolar position and root diameters of the maxillary central and lateral incisors.
Point 3: Similarly, in the discussion section the authors have limited to mention the fact that cephalometric tracing is not the suitable choice for alveolar bone evaluation in orthodontic treatment. Again, a wide reasoning about the consequences of their findings in terms of orthodontic treatment decision and planning (extraction/non extraction, the usage of inter-maxillary elastics etc) is expected.
Response 3: Thank you very much for your suggestions. We agree with the reviewer that a wide reasoning concerning the consequences of the findings in terms of orthodontic treatment decision and planning should be mentioned in the discussion section. Therefore, we added content concerning the consequences of the findings in terms of orthodontic treatment decision and planning in the discussion section.
Previous content:
Discussion Paragraph 8th Line 2;
This study provides comprehensive measurements of maxillary alveolar bone compositions of CBCT images compared to routine lateral cephalograms. This information implies that cephalometric tracing is not the suitable choice for alveolar bone evaluation in orthodontic treatment.
Amended content:
Discussion Paragraph 8th Line 2;
This study provides comprehensive measurements of maxillary alveolar bone compositions of CBCT images compared to routine lateral cephalograms. A lateral cephalogram is the standard tool for orthodontic treatment decision and planning in terms of dentoalveolar position, esthetic evaluation, and treatment options (extraction/ non-extraction/ orthognathic surgery) [23]. However, the differences observed between Ceph and CBCT in each measurement came from several factors such as radiographic magnification, anatomical geometry of proximal bone and anterior nasal spine, and errors in root tracing. Therefore, cephalometric tracing is not the suitable for alveolar bone evaluation in orthodontic treatment.
Point 4: Why did the authors do not consider to investigate mandibular incisors area? This region is even more important for orthodontic purposes, even considering the periodontal sequalae of orthodontic treatment. At least, Authors should mention this aspect. It should be interesting to investigate the ceph/CBCT differences in detecting evaluate dentoalveolar position and root diameters of the maxillary incisors (even mandibular) in different craniofacial skeletal pattern reporting (Lo Giudice A, Rustico L, Caprioglio A, Migliorati M, Nucera R. Evaluation of condylar cortical bone thickness in patient groups with different vertical facial dimensions using cone-beam computed tomography. Odontology 2020;108(4):669-675; Lo Giudice A, Caccianiga G, Crimi S, Cavallini C, Leonardi R. Frequency and type of ponticulus posticus in a longitudinal sample of nonorthodontically treated patients: relationship with gender, age, skeletal maturity, and skeletal malocclusion. Oral Surg Oral Med Oral Pathol Oral Radiol. 2018 Sep;126(3):291-297)
Response 4: Thank you very much for your concern. We agree with the reviewer that the mandibular incisors area is very important for orthodontic purposes, even considering the periodontal sequalae of orthodontic treatment. Therefore, we mentioned this aspect in the last paragraph of the Discussion. Furthermore, a comparison between Ceph and CBCT in different craniofacial skeletal patterns should be further investigated. Therefore, we have added this aspect and the suggested references in the last paragraph of the Discussion.
Previous content: -
Amended content:
Discussion Last Paragraph;
The results of this study may be applied only in the maxillary incisors. Further investigations are recommended into the differences of cephalograms and CBCT to detect and evaluate dentoalveolar positions and root diameters of all dentition areas. In particular, the mandibular incisors are important for orthodontic tooth movement, and periodontal sequalae are prone to occur following orthodontic treatment in different types of craniofacial skeletal patterns [24,25].
Point 5: In Figure 2 seems that the sagittal axis (green) is not centered and not parallel to the long axis of the tooth, while for this kind of measurements it is necessary to have consistency among measurements.
Response 5: Thank you very much for mentioning this point. We agree with the reviewer that the sagittal axis in Figure 2 is not centered and not parallel to the long axis of the tooth, which can affect the measurements. Therefore, we amended the orientation plane of Figure 2 to be in the center and parallel to the long axis of the tooth.
Previous content: Figure 2
Amended content: New Figure 2

Reviewer 2 Report
The manuscript titled “The Accuracy of Lateral Cephalogram in Representing 2 the Anterior Maxillary Dentoalveolar Position” is a very interesting topic for the clinical practice and the scientific basis of orthodontics. This is well written; however, some minor changes would improve the article.
In abstract,
The result section must be reported by numbers and p-value.
In the main body,
The ethical number of study must be added to the material and method
The reason for the difference between each measurement in Ceph and CBCT must be discussed more in the discussion section.
The errors in routine cephalometric tracing must be clearly illustrated in the discussion.
For comparison of changes of upper incisor in other methods, the following article must be cited
The effect of Alexander, Gianelli, Roth, and MBT bracket systems on anterior retraction: a 3-dimensional finite element study. Clinical Oral Investigations 24 (3), 1351-1357
Author Response
Response to Reviewer 2 Comments
Point 1: The manuscript titled “The Accuracy of Lateral Cephalogram in Representing 2 the Anterior Maxillary Dentoalveolar Position” is a very interesting topic for the clinical practice and the scientific basis of orthodontics. This is well written; however, some minor changes would improve the article.
Response 1: Thank you very much for your valuable comments. We have made changes following your suggestions.
Point 2: In abstract, The result section must be reported by numbers and p-value.
Response 2: Thank you very much for your suggestion. We added values and p-values in the Results of abstract.
Previous content:
Abstract;
Results: The cephalograms presented thicker alveolar and cortical bone as well as higher alveolar crest and cortical height than the CBCT images on both the labial and palatal sides, while cancellous thickness was not significantly greater. Moreover, the cephalograms presented a greater total root-bone thickness, whereas the cephalograms traced a thinner root than the CBCT images.
Amended content:
Abstract;
Results: The cephalograms presented thicker alveolar bone (labial: 0.20–0.67 mm, palatal: 0.41–0.60 mm, p < 0.001) and cortical bone (labial: 0.20–0.67 mm, palatal: 0.41–0.52 mm; p < 0.001) as well as higher alveolar crest (labial: 0.23–0.27 mm, palatal: 0.15–0.17 mm; p < 0.001) and cortical height (labial: 0.35–0.47 mm; p = 0.051, palatal: 0.14–0.18 mm; p < 0.001) than the CBCT images on both the labial and palatal sides, while palatal cancellous thickness was not significantly greater (p > 0.01). Moreover, the cephalograms presented a greater total root-bone thickness (0.80–1.08 mm; p < 0.001) whereas the cephalograms traced a thinner root than the CBCT images (0.36–0.52 mm; p < 0.01).
Point 3: In the main body, The ethical number of study must be added to the material and method.
Response 3: Thank you very much for your suggestion. We added the IRB number of the study (protocol EC6305-016) in the Materials and Methods.
Materials and Methods (Paragraph 1 Line 4);
2.1 Subject selection
The study protocol was approved by the Institutional Review Board for human patients (protocol EC6305-016) of the Faculty of Dentistry, Prince of Songkla University.
Point 4: The reason for the difference between each measurement in Ceph and CBCT must be discussed more in the discussion section.
Response 4: Thank you very much for your suggestion. We agree with the reviewer that the difference between each measurement in Ceph and CBCT should be discussed further. Therefore, we added this aspect in the 8th paragraph of the Discussion.
Previous content:
Discussion Paragraph 8th;
This study provides comprehensive measurements of maxillary alveolar bone compositions of CBCT images compared to routine lateral cephalograms. This information implies that cephalometric tracing is not the suitable choice for alveolar bone evaluation in orthodontic treatment.
Amended content:
Discussion Paragraph 8th;
This study provides comprehensive measurements of maxillary alveolar bone compositions of CBCT images compared to routine lateral cephalograms. A lateral cephalogram is the standard tool for orthodontic treatment decisions and planning in terms of dentoalveolar position, esthetic evaluation, and treatment options (extraction /non-extraction / orthognathic surgery) [23]. However, the differences observed between Ceph and CBCT in each measurement came from several factors such as the radiographic magnification, anatomical geometry of proximal bone and anterior nasal spine, and errors in root tracing. Therefore, cephalometric tracing is not suitable for alveolar bone evaluation in orthodontic treatment.
Point 5: The errors in routine cephalometric tracing must be clearly illustrated in the discussion.
Response 5: Thank you very much for your comment. We agree with the reviewer that the errors in routine cephalometric tracing must be clearly illustrated in the Discussion. Therefore, we added this aspect in the 9th paragraph of the discussion section.
Previous content:
Discussion Paragraph 9th;
The accuracy of lateral cephalogram in representing the anterior maxillary dentoalveolar position is the differences between cephalogram and CBCT in bone thicknesses by 0.5 mm and the bone heights by 0.2 to 0.3 mm as well as the root thickness by 0.5 mm.
Amended content:
Discussion Paragraph 9th;
Errors in routine cephalometric tracing can come from both obtaining the cephalogram (due to radiographic enlargement or distortion) and landmark identification from poor image quality or lack of experience. The accuracy or error of a lateral cephalogram in representing the anterior maxillary dentoalveolar position in this study was observed in the different values between the measurements obtained from the cephalograms and CBCT images. The error in the lateral cephalogram in detecting bone thickness was 0.5 mm while errors in bone heights were 0.2–0.3 mm, and the error in root thickness was 0.5 mm.
Point 6: For comparison of changes of upper incisor in other methods, the following article must be cited. The effect of Alexander, Gianelli, Roth, and MBT bracket systems on anterior retraction: a 3-dimensional finite element study. Clinical Oral Investigations 24 (3), 1351-1357
Response 6: Thank you very much for your suggestion. We added this reference in last sentence of the 5th paragraph of the discussion section.
Previous content:
Discussion Paragraph 5th;
Although the differences of 0.2 mm are not clinically significant, this information concealed the actual bone height which orthodontists need to be aware of when retraction of the incisors is planned.
Amended content:
Discussion Paragraph 5th;
Although differences of 0.2 mm are not clinically significant, this information concealed the actual bone height that orthodontists need to be aware of when upper incisor retraction is planned. Furthermore, the amount of force and the bracket system affect incisor and root movement [19].

Round 2
Reviewer 1 Report
The authors have successfully improved the quality of the manuscript according to my previous suggestions.
the manuscript can be published in its current form